

# Amlodipine and lufenuron as repurposing drugs against *Sporothrix brasiliensis*

Vanice Rodrigues Poester[1,2], Jéssica Estefania Dávila Hidalgo[2], Lara Severo Jardim[2], Mariana Rodrigues Trápaga[1,2], Vanessa Brito de Souza Rabello[3], Rodrigo Almeida-Paes[3], Rosely Maria Zancope-Oliveira[3] and Melissa Orzechowski Xavier[1,2]

[1] Programa de Pós-Graduação em Ciências da Saúde, Faculdade de Medicina (FAMED), Universidade Federal do Rio Grande (FURG), Rio Grande, Rio Grande do Sul, Brazil
[2] Laboratório de Micologia (FAMED), Universidade Federal do Rio Grande (FURG), Rio Grande, Rio Grande do Sul, Brazil
[3] Laboratório de Micologia, Instituto Nacional de Infectologia Evandro Chagas, Fundação Oswaldo Cruz, Rio de Janeiro, Rio de Janeiro state, Brazil

## ABSTRACT

**Background.** Sporotrichosis caused by *Sporothrix brasiliensis* is a globally emerging infectious disease with limited therapeutic options. Thus, we aimed to evaluate the *in vitro* activity of amlodipine (AML) and lufenuron (LUF) alone and their interaction with itraconazole (ITZ), the first-choice drug against *S. brasiliensis*.

**Methods.** Twenty clinical isolates of *S. brasiliensis* from two hyperendemic regions were tested through a microdilution assay to evaluate the minimal inhibitory concentration (MIC) and minimal fungicidal concentration (MFC) of AML and LUF. Checkerboard assay was performed with 10 isolates for both drug interactions with ITZ.

**Results.** AML showed inhibitory and fungicidal activity against all isolates included, with MIC values ranging from 32 to 256 µg/mL, and MFC from 64 to 256 µg/mL. However, none of the *S. brasiliensis* isolates were inhibited by the highest soluble concentration of LUF (MIC >64 µg/mL for all strains). Synergic interaction of AML and LUF with ITZ occurred in 50% and 40% of the isolates tested, without any antagonistic effects.

**Conclusion.** Both repurposing drugs evaluated in our study showed a promising *in vitro* activity, especially in synergy with ITZ against *S. brasiliensis*, warranting future *in vivo* investigations regarding its activity.

## INTRODUCTION

Sporotrichosis caused by *Sporothrix brasiliensis* poses a severe public health challenge in Brazil and had spread to other countries (Paraguay, Chile, Argentina, the United Kingdom, and the United States) over the last decade, emerging as a global infectious disease (*Rabello et al., 2022*; *Gómez-Gaviria, Martínez-Álvarez & Mora-Montes, 2023*; *Xavier et al., 2023*). Distinct genotypic profiles were shown to coexist in the major Brazilian hyperendemic areas, with *S. brasiliensis* isolates from Rio de Janeiro state (RJ) differing from those from Rio Grande do Sul state (RS) (*Rodrigues et al., 2013*; *Losada et al., 2023*; *Spruijtenburg et al.,*

Corresponding authors
Vanice Rodrigues Poester, vanicerp@gmail.com
Melissa Orzechowski Xavier, melissaxavierfurg@gmail.com

*2023*). Genotyping studies including strains from outside Brazil are not yet available in the scientific literature.

An essential aspect for controlling cases of sporotrichosis is the treatment of different hosts, especially infected cats. These animals often develop a severe and fatal form (disseminated or extracutaneous) of the disease, if they do not have access to early and effective treatment. Furthermore, cats have a high fungal load in their lesions, which increases their potential for transmitting *S. brasiliensis* to other cats and also to humans. Since the arsenal of approved drugs to treat sporotrichosis (human and feline) is limited to four antifungals (itraconazole, terbinafine, potassium iodide and amphotericin B) that are associated with many adverse effects, drug repurposing is a promising field of study. The evaluation of new antifungal compounds for this mycosis requires considering distinct genotypic profiles of *S. brasiliensis*, to account for potential variations in the susceptibility of these different strains (*De Souza et al., 2018*; *Poester et al., 2022*).

Amlodipine is a calcium channel blocker drug known to inhibit efflux pumps, which is an interesting mechanism of action against fungi and other microorganisms (*Coelho et al., 2015*; *Homa et al., 2017*; *Nakasu et al., 2021*). Regarding antifungal activity of this drug, amlodipine in combination with the salt besylate showed promising results in inhibiting and killing *C. albicans* and *C. glabrata* (minimal Inhibitory concentration, MIC; minimal fungicidal concentration, MFC values ranging from 8 to 512 μg/mL), and also showing activity in inhibiting the biofilm formation of these fungal pathogens (*Gupta et al., 2016*). Thus, studies that evaluate the activity of this drug against *S. brasiliensis*, both alone and in combination with itraconazole (ITZ), which is first-choice drug for sporotrichosis are requested. Similarly, lufenuron, an anti-ectoparasite drug that acts on chitin, presents potential as a topical treatment for *S. brasiliensis*, as chitin is an important component of fungal cell wall. In addition, this drug has been pointed as a compound to treat dermatophyte infections in animals, demonstrating its application as an antifungal therapy (*Moriello, 2004*; *Rust, 2005*). Pre-clinical studies with these both drugs against *S. brasiliensis* are not described, therefore, our study aims to evaluate their *in vitro* activity, alone and in combination with ITZ, against *S. brasiliensis* strains from two different genotypes.

## MATERIALS & METHODS

Twenty isolates of *S. brasiliensis* were included in the study, with nine originating from RJ, 10 from RS, and type strain (CBS 120339) also isolated in RJ. Isolates were obtained from clinical samples from human and feline sporotrichosis cases ($n = 18$) or from the environment ($n = 1$). All isolates were stored in the mycological collections from the participants laboratories (Mycology Laboratory from *Universidade Federal do Rio Grande*—FURG and Mycology Laboratory from *Instituto Nacional de Infectologia Evandro Chagas - Fundação Oswaldo Cruz*—Fiocruz). They have been previously identified by a species-specific polymerase chain reaction (PCR) (*Rodrigues, Hoog & Camargo, 2015*). To evaluate the genotype of isolates, eight strains were genotyped by partial sequences of the translation elongation factor-1 alpha (EF1$\alpha$) and the calmodulin gene (CAL), following the PCR conditions described by *Marimon et al. (2007)* and *Rodrigues et al.*

*(2013)*, respectively. Automated sequencing was done using the FIOCRUZ Technological Platforms and the sequences were edited with the Sequencher software package (version 4.9; (Gene Codes Corporation, Ann Arbor, MI, USA)). Phylogenetic analyses were carried out using maximum likelihood method, and trees were constructed using MEGA 6 (*Tamura et al., 2013*), confidence values were performed using 1000 bootstrap replicates and they were shown next to the branches (*Felsenstein, 1985*). *S. brasiliensis* sequence from this study was deposited at GenBank (numbers: OQ865503, OQ865516, KC576606, AM116899, OQ865505, OQ865518, OQ865506, OQ865519, OQ865507, OQ865520, OQ865508, OQ865521, OQ865509, OQ865522, OQ865510, OQ865523, KC576614, AM117437, KC576608, AM116908, KC576615, AM747302, KC576611, AM398393, KC576612, AM398396, MW066427, MW075142) sequences belonging to the others *Sporothrix* species deposited at GenBank were included in the phylogenetic analysis and *Ophiostoma pallidulum* was used as outgroup. The haplotype network was built with the software Network 10.2.0.0 using the Median-joining networks method (*Polzin & Daneschmand, 2003*), gaps and missing data were excluded from the analysis.

Drugs were obtained commercially and include ITZ (Sigma-Aldrich®, San Luis, Missouri, EUA), amlodipine (Valdequimica®, São Paulo, Brazil), and lufenuron (Copervet®, Minas Gerais, Brazil). These drugs were diluted and stored as stock solutions in dimethyl sulfoxide, 51.200 µg/mL to amlodipine and 6.400 µg/mL to lufenuron and ITZ.

The *in vitro* activities of drugs were evaluated through the microdilution assay, following the M38-A2 protocol from the Clinical and Laboratory Standards Institute (CLSI, 2008). The solubility of lufenuron and amlodipine in RPMI 1640 medium was tested to define their highest testable concentration, resulting in a range of 1 to 64 µg/mL and 8 to 512 µg/mL, respectively. DMSO maximum concentration in the well of susceptibility test was 1% (CLSI, 2008). Isolates from seven days on potato dextrose agar (PDA) (Kasvi®, São José dos Pinhais, Paraná, Brazil) with their concentration adjusted to $0.8 \times 10^4$ to $10^5$ colony-forming units (CFU) per mL by spectrophotometry (530 nm). To confirm the inoculum concentration, the pour-plate technique was performed and colonies were counted after seven days of incubation. A standardized solution of inoculum and drug stock solutions were diluted in RPMI 1640 medium and distributed into 96-well polystyrene plates (100 µl of inoculum and 100 µl of drug dilutions). The microplates were then incubated for 72 h at 35 °C. Visual readings were made to determine the MIC of each drug, defined as the concentration that completely inhibited fungal growth. In addition, the MFC was evaluated through plating 50 µl of each well without visual growth on PDA. The MIC/MFC50, MIC/MFC90 (concentration able to inhibit/kill 50 and 90% of the isolates, respectively), and geometrical means (GM) were calculated.

Ten isolates of the twenty (four from RJ, five from RS—randomly selected, and the *S. brasiliensis* type strain) were used for drug interaction evaluation (amlodipine + ITZ or lufenuron + ITZ) by a checkerboard assay (*Eliopoulos & Moellering, 1991*; *Poester et al., 2020*). The concentrations of repurposing drugs and test conditions were performed as described above, and ITZ was tested in concentrations from 0.03125 to 8 µg/mL. ITZ MIC values were classified as wild-type (<2 µg/ml) or non-wild-type ($\geq$ 2 µg/ml) using

the epidemiological cutoff values (ECVs) described by *Espinel-Ingroff et al. (2017)*. In the checkerboard assay, the activity of drugs are tested alone (ITZ and amlodipine or lufenuron) and in combination, in the same concentrations described above, to include values in the equation: (MICa in combination/MICa tested alone) + (MICb in combination / MICb tested alone), being MICa: amlodipine or lufenuron and MICb: ITZ. The equation determined the fractional inhibitory concentration index (FICi) used to classify the drug associations as follows: strong synergism (SS) when FICi <0.5, weak synergism (WS) when 0.5 <FICi <1, additive (AD) when 1 <FICI <2, indifferent (IND) when FICi = 2, and antagonistic (ANT) when FICi >2 (*Poester et al., 2020*).

## RESULTS

Eight *S. brasiliensis* isolates included genotype analyses were separated into two distinct groups, according to the haplotype network constructed using the concatenated $EF1\alpha$ and CAL sequences. These genotypes separated isolates from RJ (genotype H1) and RS (genotype H2 and H3) (Fig. 1).

Figure 2 summarizes the MIC results of the three drugs herein tested. In brief, ITZ exhibited MIC values ranging from 0.125 to 1 μg/mL (MIC50 and MIC90 of 1 μg/mL), The GM of ITZ MIC values for RJ and RS isolates was 1 and 1.19 μg/mL, respectively. Additionally, two isolates (one from RJ and the other from RS) were classified as non-wild type, showing MIC values >8 μg/mL for this azole (Fig. 2).

Amlodipine showed both inhibitory and fungicidal activity against all isolates ($n = 20$) with MIC values ranging from 32 to 256 μg/mL (MIC50 and MIC90 of 128 μg/mL). The GM of MIC values for RJ and RS isolates was 97 and 103.97 μg/mL respectively. MFC values ranged from 64 to 256 μg/mL (MFC50 of 128 μg/mL, and MFC90 of 256 μg/mL), GM of 194.01 μg/mL and 157.59 μg/mL to RJ and RS isolates, respectively. In contrast, lufenuron did not inhibit any *S. brasiliensis* isolates, with MIC values higher than 64 μg/mL for all strains.

Regarding the interaction of drugs with ITZ, when in association with amlodipine, a beneficial interaction was observed in 60% of cases (10% SS, 40% WS, and 10% AD), while 40% showed indifference. In association with lufenuron, 40% WS was found, and 60% of isolated showed indifference (Table 1).

## DISCUSSION

Our study showed the *in vitro* activity of two repurposing drugs, either alone or in association with ITZ, against the pathogenic species *S. brasiliensis*. These isolates were obtained from clinical cases in the two main sporotrichosis hyperendemic regions in Brazil (*Gremião et al., 2020*; *Munhoz et al., 2022*; *Losada et al., 2023*; *Spruijtenburg et al., 2023*). Sporotrichosis represents a severe public health problem in Brazil and currently it is emerging as a global concern. Repurposing drugs is a promising strategy for investigating new potential antifungals against *Sporothrix* spp., since their pharmacological information are already available, which reduces the time needed to develop and discover new therapies.

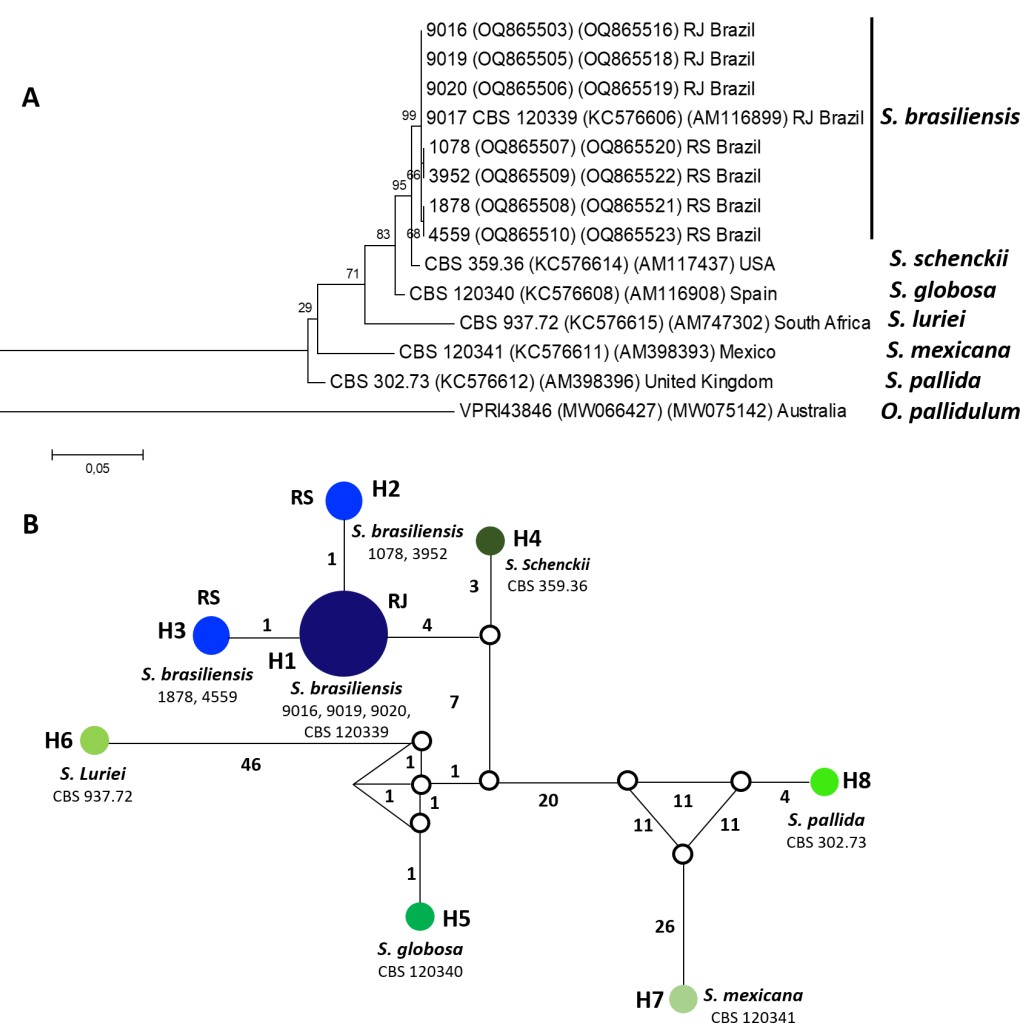

**Figure 1** Phylogenetic relationships and Haplotype network of *Sporothrix brasiliensis* from Rio de Janeiro (RJ) and Rio Grande do Sul (RS) states.

Amlodipine demonstrated inhibition and killing of all included isolates, and its activity was further increased when combined with ITZ, the drug of choice for sporotrichosis. A promising antifungal activity of amlodipine was also showed in combination with fluconazole against *C. albicans*, changing the resistance status of strains to the azole drug (*Liu et al., 2016*). Unfortunately, this was not observed with the two ITZ non-wild-strains included in this study. Regarding toxicity of amlodipine, genotoxicity was suggested, but not conclusively proven, and cytotoxicity was observed only at higher doses (204.44 μg/mL) than the MIC90 value (128 μg/mL) found in our study (*Zheng et al., 2010*; *Salih et al., 2022*).

Lufenuron, a compound proposed to treat dermatophyte infections in animals (*Moriello, 2004*), did not demonstrate inhibitory activity against *S. brasiliensis* in our study. However, its topical application may complement systemic ITZ therapy for sporotrichosis, since a beneficial effect of its *in vitro* association with ITZ was shown in our study for some isolates.

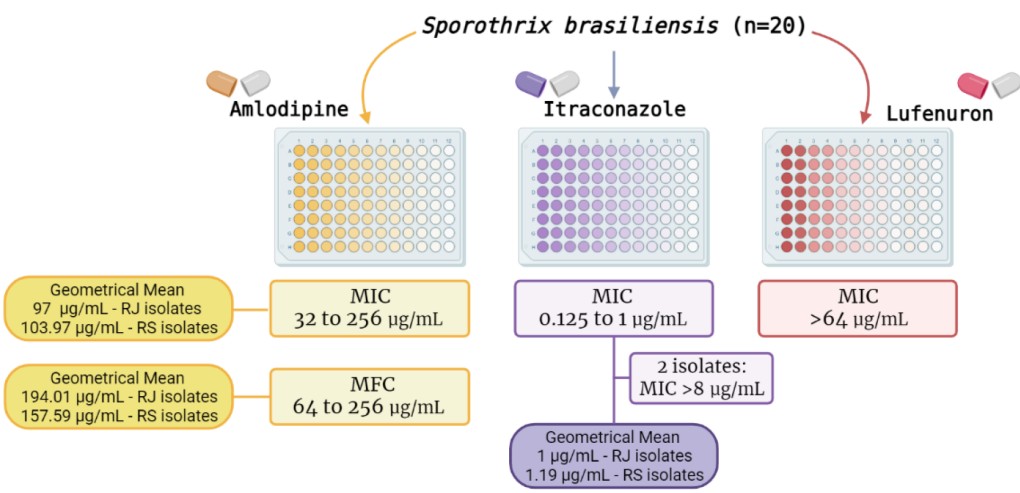

**Figure 2** Results of the *in vitro* susceptibility of 20 *Sporothrix brasiliensis* isolates from Rio de Janeiro (RJ) and Rio Grande do Sul (RS) states to amlodipine, lufenuron and itraconazole.

**Table 1** Results of the *in vitro* susceptibility of 10 *Sporothrix brasiliensis* isolates to amlodipine (AML) and lufenuron (LUF) in combination with itraconazole (ITZ).

| FURG ID | Source | MIC[a] | | | | IN[b] | MIC | | | | IN[b] |
|---|---|---|---|---|---|---|---|---|---|---|---|
| | | AML alone | AML comb | ITZ alone | ITZ comb | | LUF alone | LUF comb | ITZ alone | ITZ comb | |
| 716 | RS | 64 | 32 | 1 | 0.25 | WS | >64 | 16 | 1 | 0.5 | WS |
| 1078 | RS | 32 | 8 | 1 | 0.5 | WS | >64 | >64 | 1 | 1 | IND |
| 1878 | RS | 64 | 8 | 0.5 | 0.125 | SS | >64 | >64 | 0.5 | 0.5 | IND |
| 3952 | RS | 64 | 64 | >8 | >8 | IND | >64 | >64 | >8 | >8 | IND |
| 5150 | RS | 64 | 32 | 1 | 0.25 | WS | >64 | 16 | 1 | 0.5 | WS |
| 9011 | RJ | 64 | 64 | 0.5 | 0.5 | IND | >64 | >64 | 0.5 | 0.5 | IND |
| 9013 | RJ | 32 | 32 | 1 | 1 | IND | >64 | 4 | 1 | 0.5 | WS |
| 9014 | RJ | 128 | 128 | >8 | >8 | IND | >64 | >64 | >8 | >8 | IND |
| 9015 | RJ | 32 | 8 | 0.5 | 0.25 | WS | >64 | 1 | 0.5 | 0.25 | WS |
| 9017 | ATCC | 64 | 32 | 1 | 0.5 | AD | >64 | >64 | 1 | 1 | IND |

**Notes.**

FURG ID, Isolate identification of *Universidade Federal do Rio Grande*; MIC, Minimal inhibitory concentration; comb, MIC of each drug when used in combination; IN, Interpretation.

[a]MIC expressed as μg/mL.

[b]IN: <0.5 strong synergism (SS); 0.5–<1 weak synergism (WS); 1–<2 additive (AD); 2 indifferent (IND); >2 antagonism (AN).

While our study showed a similar susceptibility profile to amlodipine, lufenuron, and ITZ between the RJ and RS isolates, it is important to highlight the necessity to include genotypically diverse *Sporothrix* isolates in all studies aiming to discover new antifungal drugs as well as to test the susceptibility for commercial approved drugs. In fact, it is suggested that isolates from RJ could be less resistant than those from RS, since 85% of patients from the RJ hyperendemic area acquired clinical cure using ITZ 100 mg/day, on the other hand 100% of those from RS needed to increase doses from 100 to 200 or 400 mg/day

during the treatment (*Barros et al., 2011*; *Poester et al., 2022*). The isolates tested in our study were originated from these two epidemiological sources (RJ and RS) of the Brazilian hyperendemic, which probably underwent clonal dispersion to other states (*Losada et al., 2023*; *Spruijtenburg et al., 2023*).

## CONCLUSIONS

Given the urgent need for more therapeutic options to control the high dissemination of sporotrichosis, our study is pioneering in showing the activity of two repurposing drugs alone and/or in association with ITZ against *S. brasiliensis* from two epidemiological sources in Brazil. The drugs evaluated are promising as future antifungals that would contribute to animal and human sporotrichosis treatment. Therefore, our preliminary *in vitro* results instigate further pre-clinical studies (both *in vitro* and *in vivo*) with both repurposing drugs herein evaluated. These studies hold the potential to advance the development of new treatment strategies for this challenging infectious disease.

## ACKNOWLEDGEMENTS

The authors are grateful to *Coordenação de Aperfeiçoamento de Pessoal de Nível Superior* (CAPES). The contents of this manuscript are solely the responsibility of the authors and do not necessarily represent the official views of the National Institutes of Health.

### Funding
This work was supported by the Conselho Nacional de Desenvolvimento Científico e Tecnológico (CNPq, numbers 405653/2021-2, 308315/2021-9 and 316067/2021-0), Programa Inova Fiocruz (number PRES-008-FIO-22-2-11), Fundação Carlos Chagas Filho de Amparo à Pesquisa do Estado do Rio de Janeiro (FAPERJ, numbers: E-26/201.441/2021 and E-26/200.381/2023) and Fundação de Amparo à pesquisa do Estado do Rio Grande do Sul (FAPERGS, number: 21/2551-0001974-3). The funders had no role in study design, data collection and analysis, decision to publish, or preparation of the manuscript.

### Grant Disclosures
The following grant information was disclosed by the authors:
Conselho Nacional de Desenvolvimento Científico e Tecnológico: 405653/2021-2, 308315/2021-9.
Programa Inova Fiocruz: PRES-008-FIO-22-2-11.
Fundação Carlos Chagas Filho de Amparo à Pesquisa do Estado do Rio de Janeiro: E-26/201.441/2021, E-26/200.381/2023.
Fundação de Amparo à pesquisa do Estado do Rio Grande do Sul: 21/2551-0001974-3.

### Competing Interests
The authors declare there are no competing interests.

## Author Contributions

- Vanice Rodrigues Poester conceived and designed the experiments, performed the experiments, analyzed the data, prepared figures and/or tables, authored or reviewed drafts of the article, and approved the final draft.
- Jéssica Estefania Dávila Hidalgo performed the experiments, prepared figures and/or tables, and approved the final draft.
- Lara Severo Jardim performed the experiments, prepared figures and/or tables, and approved the final draft.
- Mariana Rodrigues Trápaga performed the experiments, analyzed the data, prepared figures and/or tables, and approved the final draft.
- Vanessa Brito de Souza Rabello performed the experiments, analyzed the data, prepared figures and/or tables, and approved the final draft.
- Rodrigo Almeida-Paes conceived and designed the experiments, analyzed the data, authored or reviewed drafts of the article, and approved the final draft.
- Rosely Maria Zancope-Oliveira conceived and designed the experiments, analyzed the data, authored or reviewed drafts of the article, and approved the final draft.
- Melissa Orzechowski Xavier conceived and designed the experiments, analyzed the data, authored or reviewed drafts of the article, and approved the final draft.

## Data Availability

The sequences are available at Genbank: OQ865503, OQ865516, KC576606, AM116899, OQ865505, OQ865518, OQ865506, OQ865519, OQ865507, OQ865520, OQ865508, OQ865521, OQ865509, OQ865522, OQ865510, OQ865523, KC576614, AM117437, KC576608, AM116908, KC576615, AM747302, KC576611, AM398393, KC576612, AM398396, MW066427, MW075142.

## Supplemental Information

Supplemental information for this article can be found online at http://dx.doi.org/10.7717/peerj.16443#supplemental-information.

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
