# Peer review of "Amlodipine and lufenuron as repurposing drugs against Sporothrix brasiliensis"

_PeerJ, doi:10.7717/peerj.16443_

## Round 0.1 · original submission · Minor Revisions

All three authors seemed positive about the importance of this work, given the spread of Sporothrix brasiliensis. Most comments were relatively minor.

Reviewer 1 ·

Basic reporting

The manuscript communicates an interesting and relevant subject that is a priority in medical mycology. Since Sporothrix brasiliensis is the causative agent of an epidemic outbreak now affecting South America, the manuscript content is relevant and timely. The experimental design supports the results and the conclusion is based on the reported observations. My only concern is the lack of in vivo data to support the in vitro observations. As a consequence, the study seems to be in a preliminary stage that requires in vivo confirmation. Currently, with the diverse set of models of experimental sprotrichosis this seems feasible in the short term. The authors are encouraged to include this relevant and key aspect.
As a minor issue, it is not described whether the excipient where the drugs were included had a role in the observed effects, nor the way that DMSO affected fungal cells.

Experimental design

.

Validity of the findings

.

·

Basic reporting

The English language is good, however, the writing needs to be improved in several paragraphs of the text (this is noted in the PDF file). The bibliographic references are related to the research topic, it is necessary to review the format of the bibliography (pages 11 and 12). The article has an orderly structure and the most relevant points are mentioned in each section. The legends of the tables and figures are fine, only some spelling errors need to be corrected. The figures are interesting and easy for the reader to understand.

Experimental design

I consider this work to be novel, taking into account the rapid expansion of S. brasiliensis in Brazilian territory. Being an “endemic” species, its treatment has become increasingly difficult. In general, the design of the experiments is planned correctly, however, I consider it necessary that the materials and methods section specify where the samples were obtained from. It would also be interesting if they tried to replicate the experiments using other culture media, and to perform cytotoxicity tests for both drugs.

Validity of the findings

The methodology used contributes to the work being complete. The statistical analyzes used and the bioinformatics tools are indicated to answer your research question. The conclusions are related to the work and have future perspectives in this area of research. I only consider it important to take into account some other experiments as I mentioned before, so that the work is more complete.

Additional comments

I consider it important to emphasize more in the introduction why they used those two drugs. It should also be mentioned in the text a little bit about sporotrichosis in cats, since the most common etiological agent of this zoonosis is S. brasiliensis.
Would this treatment also work for cats? It is a bit confusing as they do not specify to which organisms it could be administered. In the PDF file I suggest some corrections for the introduction and discussion.

·

Basic reporting

1. In the introduction, it's interesting that a brief mention is made of the mechanism of action of amlodipine and lufenuron. However, since they are the focus of this research, it would be good to add more information about fungal organisms in which their activity has already been evaluated. Even a demonstrative figure of their mechanisms of action would be of interest to the reader.
2. In lines 171-173, it would be good to specify which drug, when combined with fluconazole, was responsible for the antifungal activity against Candida albicans. This clarification is necessary because in the previous statement, amlodipine and itraconazole are mentioned, which could potentially confuse inexperienced readers.
3. In Table 1, the initials of amlodipine are misspelled.
4. In the MIC section, I understand that the first part is analyzing amlodipine alone, then amlodipine combined with itraconazole, and finally itraconazole alone. However, in the last column, it mentions itraconazole combined. I suggest explaining how these combinations are organized more clearly in the methodology. Since in lines 109 and 110, it mentions "were used for drug interaction evaluation (amlodipine + ITZ or lufenuron + ITZ) by a checkerboard assay," it implies that there are only three possible combinations per drug.
5. Maintain consistency in the references, especially in the provided links (some have hyperlinks, and others do not).

Experimental design

no comment

Validity of the findings

no comment

Additional comments

I appreciate that you provided the raw data; however, I suggest avoiding spaces in the sequence names. As a suggestion, you can either remove spaces or use "_" or "-" for separation.

---

## Round 0.2 · accepted · Accept

The authors appear to have adequately responded to all concerns from the reviewers, strengthening the manuscript.

One very minor suggestion if you're able to edit it at this point - Figure 1b: the l for luriei should be lowercase. Figure 1a: the scale bar should have units (could be added to the figure legend).